# Cognitive Control Architecture for the Practical Realization of UAV Collision Avoidance

**DOI:** 10.3390/s24092790

**Published:** 2024-04-27

**Authors:** Qirui Zhang, Ruixuan Wei, Songlin Huang

**Affiliations:** 1Aviation Engineering School, Air Force Engineering University, Xi’an 710038, China; 2Unit 93535 of PLA, Rikaze 857060, China

**Keywords:** unmanned aerial vehicles, cognitive control architecture, conditioned reflex, anti-collision

## Abstract

A highly intelligent system often draws lessons from the unique abilities of humans. Current humanlike models, however, mainly focus on biological behavior, and the brain functions of humans are often overlooked. By drawing inspiration from brain science, this article shows how aspects of brain processing such as sensing, preprocessing, cognition, obstacle learning, behavior, strategy learning, pre-action, and action can be melded together in a coherent manner with cognitive control architecture. This work is based on the notion that the anti-collision response is activated in sequence, which starts from obstacle sensing to action. In the process of collision avoidance, cognition and learning modules continuously control the UAV’s repertoire. Furthermore, simulated and experimental results show that the proposed architecture is effective and feasible.

## 1. Introduction

One of the key hallmark requirements of unmanned aerial vehicles (UAVs) is the ability to avoid obstacles autonomously [1,2,3,4]. Many scholars have studied automatic collision avoidance, with the aim of pushing forward the development of UAVs. Beard [5] constructed Voronoi diagrams to avoid known obstacles. But Voronoi’s method overlooked the obstacles’ effective range, as S. L. Wang [6] pointed out. Wang further developed Voronoi diagrams into Laguerre diagrams to overcome this drawback [7]. However, diagram-based methods cannot deal with pop-up obstacles [8], since those methods require all the obstacles’ information, which is almost impossible to apply in a dynamic environment. Also, many experts are engaged in making UAVs more autonomous. They have proposed many approaches in the literature, such as A-Star and D-Star [9,10,11], potential fields [12], model predictive control [13], and evolutionary algorithmic techniques [14,15].

Although many efforts have been made in collision avoidance, UAVs’ anti-collision capability remains a difficult problem for two reasons: (1) The traditional methods assume that the obstacles are few and far between each other. (2) The time consumed is limitless. So, the traditional methods may not be practical for real-time collision avoidance, but this may be solved by referring to advances in brain-inspired cognitive systems [16].

The ability to avoid obstacles quickly is easy for humans but difficult for UAVs [17]. Douglas Hofstadter envisioned this when he said that “for any program to handle letterforms with the flexibility that human beings do, it would have to possess full-scale artificial intelligence” [18]. Many researchers have conjectured that collision avoidance could be achieved by incorporating insights from brain and cognitive science research.

One may avoid crowded obstacles without any thinking. We are not born with avoidance skills, but through explicit and repetitive training, we can develop what is called an acquired conditioned reflex (CR) [19,20,21]. People can learn the skill through special training.

The foundation of artificial intelligence (AI) lies in its capacity for self-learning, a concept. This self-learning capability is not just a theoretical construct but a fundamental requirement for the development of autonomous systems. J. Weng [22], in 2001, delineated an autonomous learning mechanism that is pivotal in the construction of intelligent robots capable of functioning with a degree of autonomy [23,24]. This mechanism is designed to enable robots to learn from their interactions with the environment, thereby improving their performance over time. M. S. Lim [25] contributed to the field by proposing a rapid and efficient method for synthesizing humanoid robot hands that can adeptly grasp planar polygonal objects. This innovation is a step toward more dexterous and versatile robotic manipulation. M. Asada [26] developed a synthetic approach for cognitive robots, which was a significant step in endowing machines with cognitive abilities that were once the exclusive domain of biological systems. The approaches mentioned in [27] further build upon these ideas, expanding the scope of AI in robotics. Behavior-based control architectures, as suggested by Simmons [28], Rosenblatt, and Thorpe [29], draw inspiration from Brooks’ [30] behavior-based subsumption architecture. These architectures are designed to facilitate complex behaviors in autonomous mobile robots through a decentralized control system that allows for the integration of various behavioral modules. In 2007, Bermudez and Verschure [21] proposed a bio-inspired UAV system, marking a significant attempt to integrate biological intelligence into the automatic control of UAVs. This system aims to leverage principles from biology to enhance the adaptability, efficiency, and autonomy of UAVs, particularly in dynamic and challenging environments. The integration of these cognitive and learning principles into UAV systems is crucial for creating machines that can operate with a higher level of autonomy and intelligence. It is through this lens of cognitive learning and adaptive behavior that the future of UAV technology is being shaped, and this has the potential to revolutionize the way we approach robotics and AI.

In this paper, we introduce a cognitive control architecture for UAVs that is inspired by the human brain’s conditioned-reflex mechanisms. Our approach significantly enhances the real-time collision avoidance capabilities of UAVs by reducing their computational complexity and response time. The main contributions of this paper include the development of an innovative learning-based strategy that allows UAVs to adapt to dynamic environments and avoid obstacles more efficiently. We have conducted extensive simulations and experiments to validate the effectiveness of our proposed system, which demonstrates a substantial improvement over traditional methods in terms of both safety and operational efficiency.

## 2. A Cognitive Control Architecture for the Conditioned Reflex Cycle in the Anti-Collision Behavior of UAVs

### 2.1. The Brain Pathway of the Conditioned Reflex Cycle in Human Anti-Collision Behavior

Humans have a fundamental function, which is to find a safe path in crowded environments based on the CR. The classical CR refers to a learning process in which a potent stimulus (e.g., obstacles) is paired with a previously neutral stimulus (e.g., anti-collision movements); its neural substrates are now beginning to be understood [31,32]. The more familiar the crowded environment is, the faster and safer will be the path obtained. So, the CR-based method usually takes less time than normal ways. 

For complex problems, humans can find optimal solutions through cognition and learning [33,34]. Infants may not know how to avoid obstacles in the complex environment, but they can cognize the obstacles and learn how to deal with obstacles as they grow up. This process is regarded as a growth of the brain, especially the parts concerning anti-collision behavior. There are three important parts when humans are performing collision avoidance, which are the cortex, the thalamus and the basal ganglia. In the mammalian brain, feedback connections between brain areas play important roles in collision avoidance [35]. Rivara [36] came up with the study that suggested that the cerebral cortex has many functions, including movement planning, behavior control, and motion execution.

The thalamus has multiple functions, and it is generally believed to act as a relay station or hub, relaying information between different subcortical areas and the cerebral cortex [37]. The thalamus receives signals from cortex and produces awareness of the obstacles [38].

Popular theories implicate the basal ganglia primarily in action selection—in helping to decide which of several possible behaviors to execute [39]. In more specific terms, the basal ganglia’s primary function is likely to control and regulate activities of the motor and premotor cortical areas so that voluntary movements can be performed smoothly [40]. In conclusion, the basal ganglia is strongly interconnected with the cortex and thalamus, and its function is to control movements [41]. An anti-collision mechanism in humans can be derived based on the studies of brain science, as shown in Figure 1.

In Figure 1, the anti-collision of normal method and CR method is presented with black flow and red flow respectively. 

To normal anti-collision reflex (see in Figure 1, the black flow), there are 8 procedures in fulfilling complete anti-collision. (1). Sense. Human must have a way, see or touch, to sense the environment. (2). Preprocess. The prefrontal cortex receives and synthesizes information from the outside world [42]. The perceived information is then sent to frontal cortex, functioning on transforming original visual information to classified and reduced signals [43]. This preprocessing towards information makes subsequent steps more effective. (3). Cognize. In mammalian brain anatomy, the thalamus is the large mass of gray matter in the dorsal part of the diencephalon of the brain with several functions such as relaying of sensory signals, including motor signals, to the cerebral cortex, and the regulation of consciousness and alertness [38]. The function of thalamus, regarding anti-collision, is to classify obstacles and to have a consciousness of familiar environment, that is to cognize obstacles. (4). Obstacles learning. The cortex, in medicine, is the cerebral cortex which covers the brain, purposing on learning and reserving knowledge, and mapping the knowledge between each other [44], in other word, to develop brain through learning and to think a way out in crowded obstacles. (5). Behave. The basal ganglia is in the brains of vertebrates including humans, which is situated at the base of the forebrain. The basal ganglia is associated with a variety of functions including: control of voluntary motor movements, procedural learning, routine behaviors and emotion [41]. Its function is to design anti-collision strategy and behave safely in the environment. (6). Strategies learning. The function of frontal cortex is to memorize anti-collision strategies, and match them with obstacles. (7). Pre-act. The anti-collision strategies cannot be decoded and acted by limbs, so they should be clarified so as to transform the strategies to body language. (8). Act. The limbs follow the commands from brain to avoid crowded obstacles. 

To CR-based anti-collision (see in Figure 1, the red flow): Conditioned reflex occurs when a conditioned stimulus is paired with an unconditioned stimulus. After the pairing is repeated, the organism exhibits a conditioned response to the conditioned stimulus when the conditioned stimulus is presented along, and it must be acquired through experience. According to Bouton [45], conditioned reflex does not involve the acquisition of any new behavior, but rather the tendency to respond in old ways to new stimuli. In conditioned reflex anti-collision, there are only two processes, which are sense (9) and act (10). Humans can avoid obstacles by body language through training and experience. In a crowded environment, humans do not have any time to go through the normal anti-collision process. To make themselves safe, they can only move without thinking. But the conditioned reflex is based on normal anti-collision behavior.

### 2.2. Cognitive Control Architecture for UAVs

In collision avoidance, traditional UAVs only have two levels (a guidance level and a motion level) to plan a safe path. However, the guiding algorithms always stay the same. Even in the most familiar environment, the UAV must calculate every path from scratch.

In a cognitive UAV, there are three levels (see Figure 2; cognition level, guidance level, and motion level). The flight control strategy is executed by the guidance level, which generates control variables of the UAVs. The motion level responds to the guidance level, which transforms the control variables into throttle commands, which regulate the rotational speed of each motor, so as to control the altitude and attitude. What differs from traditional UAVs is the cognition level, which makes a UAV learn what obstacles are and how to avoid obstacles without much thinking in familiar environments.

As shown in Figure 2, information about obstacles is transferred to the cognition controller so as to produce avoidance strategies. The strategies are sent to the guidance controller to calculate and generate flight instructions. Finally, the motion controller decodes flight instructions into flying motions that directly control a UAV. In this paper, we assume that a cognitive UAV can decode avoidance strategies and obtain flying motions by itself.

### 2.3. The Brain-Like Conditioned Reflex Cycle of Cognitive UAVs

Building models with reference to the mammalian brain is an important area of study for UAVs [46,47]. Collision avoidance in a complex environment has met with tremendous difficulties for it requires the following cognitive characters:Universality: Cognitive algorithms design a universal schema rather than preprogramming and reprogramming for special missions, so they can enrich themselves incrementally.Interactivity: Humans act as teachers rather than programmers. We can only influence the process and the context but are definitely not the decision-makers.Learnability: Cognition is a gradual and cumulative process. Advanced knowledge relies on basic ability.

We introduce a hierarchical model called the recursive brain network (RBN) that incorporates brain-science studies to fulfill the cognitive demands. In RBN, the anti-collision process can be described as shown in Figure 3.

In Figure 3, the intricate flow processes that govern anti-collision systems in UAVs differ from those in traditional anti-collision methods and those inspired by conditioned reflexes. The black curves delineate the multi-step normal anti-collision procedures, which consist of eight distinct processes. These processes are designed to systematically guide the UAV through obstacle detection, analysis, and evasion, ensuring safe navigation. As indicated in the figure, each process is crucial for the UAV’s cognitive control architecture, starting from the initial sensing of the environment to the final execution of avoidance maneuvers.

The first process, ”Sensing“, involves the utilization of the UAV’s sensors to detect and identify obstacles in the vicinity. This is followed by ”Preprocessing”, where the raw sensor data is refined and organized into a usable format. ”Cognition” then interprets this information, allowing the UAV to understand and classify the obstacles within its operational space. ”Obstacle learning” is where the UAV retains and recalls information about the environment, creating a knowledge base for future navigation.

The subsequent processes, ”Behavior” and ”Strategy learning”, are responsible for developing and implementing collision-avoidance tactics. ”Pre-action” serves as the preparatory stage, where the UAV formulates a response based on its learned strategies, and ”Action” is the culmination of these processes, executing the avoidance actions with precision. These eight processes, as detailed in Table 1, form a coherent and comprehensive approach to avoiding collision, ensuring that the UAV can navigate autonomously and safely in dynamic environments.

In conditioned-reflex anti-collision (see Figure 3; the red flow), there are only two main processes, which are sensing (9) and action (10). Cognitive UAVs, like humans, can avoid obstacles directly without repetitive calculations via a conditioned reflex after training.

Figure 4 of our manuscript illustrates the detailed interplay between the trust region, flight path, and waypoints, which are crucial components in the cognitive control architecture for UAVs. The trust region is a spatial area created around the UAV, delineated by the boundaries of detected obstacles, within which the UAV is deemed to operate safely. This region forms the foundation for all subsequent navigation decisions. The flight path is derived from the trust region through a series of complex calculations that consider the UAV’s current position, the geometry of the trust region, and the intended destination. These calculations are performed by sophisticated algorithms designed to generate a trajectory that maximizes safety and efficiency. However, since the flight path is an output of algorithmic processing, there might be instances where the path includes maneuvers, such as sharp turns or abrupt changes in altitude, that exceed the UAV’s performance capabilities. For example, the flight path may include right angles that the UAV cannot physically achieve without risking a stall or control loss.

To address this, the calculated flight path undergoes further refinement to ensure that it is applicable to the UAV’s actual flight dynamics. This involves smoothing out the path to remove any extreme angles and adjusting the trajectory to fit within the UAV’s operational limitations. The end result is a flight path that is not only safe but also feasible for the UAV to follow. Once the flight path is finalized, it must be translated into a series of waypoints. These waypoints are specific points along the flight path that the UAV will navigate to, one after another, to reach its destination. The conversion of the flight path into waypoints is based on the UAV’s sensing, processing, and actuation capabilities, ensuring that each waypoint is reachable and executable.

In the graphical representation of Figure 4, filled and empty circular nodes symbolize features and pools, respectively. Features represent distinct characteristics of the environment, such as the location of obstacles, while pools are collections of possible flight paths that can be taken from one feature to another. Each pool is a composite of factors from its level and serves as a hub for the subsequent level’s features. The lateral constraints, depicted as grey rectangles, play a critical role in this model. They act as coordinators, dictating the selection among the pools they are connected to, reflecting the sequential nature of the flight-path construction process.

This comprehensive approach ensures that the UAV’s flight path is not only a product of calculations based on the trust region but also a reflection of the UAV’s cognitive understanding of its environment, resulting in a flight strategy that is both intelligent and adaptable.

Upon examining the recursive brain network (RBN) model in conjunction with the mechanisms of the human brain, a striking congruence is observed. This alignment suggests that a brain-inspired anti-collision mechanism for unmanned aerial vehicles (UAVs) can be effectively engineered to mimic cognitive processes. As depicted in Figure 5, the RBN model is divided into distinct regions that correspond to various functional areas of the brain. The blue region symbolizes the cortex function, which is responsible for higher-order cognitive processes such as planning and decision-making. The yellow region represents the thalamus function, which acts as a crucial relay center that processes and integrates sensory information before it reaches the cortex. Lastly, the purple region signifies the basal ganglia function, which plays a pivotal role in selecting and initiating appropriate actions or motor responses.

This tri-colored representation in Figure 5 not only underscores the complexity of the brain’s anti-collision response but also provides a structured framework for designing UAV systems that can exhibit cognitive-like behavior. The cortex, thalamus, and basal ganglia work in concert to enable quick and informed decisions regarding obstacle avoidance, much like a human’s innate ability to navigate through cluttered spaces without explicit calculation. By emulating these brain functions, the RBN model offers a sophisticated approach to enhancing UAV autonomy, particularly in high-stake scenarios in which rapid and accurate collision avoidance is paramount.

## 3. Algorithmic Design of Cognitive UAVs

There are two main parts in fulfilling an algorithmic design of a cognitive UAV. Firstly, we should build the mapping relation between obstacles and avoidance strategies (see Figure 5; the blue part), which is the learning part of cognitive UAVs. Then the corresponding obstacle pattern and avoidance strategy should be constructed so as to cognize obstacles and strategies (see Figure 5; the yellow and purple parts).

### 3.1. Construction of the Obstacle-Avoidance Mapping Relation

Obstacle development has a one-to-one correspondence at the beginning, which means that there is only one avoidance strategy $AS_{l}$ corresponding to each obstacle pattern $OP_{k}$. However, there may be a similar obstacle pattern in the memory of the cortex region, which can replace the obstacle pattern OP that the UAV is currently facing, which gradually accumulates in obstacle learning. Equation (1) shows a mapping relation f that outputs a safe path S in the anti-collision zone Ω based on the surrounding environment.
(1)$$S\left(t+1\right)=f\left(\begin{array}{l} S\left(t\right),O\left(t\right),\\ OP\left(t\right),AS\left(t\right) \end{array}\right),O\notin \Omega ,S\in \Omega$$

An anti-collision path can be obtained via obstacle pattern mapping, but it does not mean that the obstacle-avoidance mapping relation can be constructed. Consider two obstacle patterns: OP and $OP_{3}$ (see Figure 6). The avoidance strategy $AS_{1}$ represents an anti-collision path; nevertheless, the optimal mapping relation $OP_{3}-AS_{1}$ is not optimal for the obstacle pattern OP.

Unlike a normal obstacle-avoidance mapping structure, cognitive mapping relation is an incremental learning schema. It can not only output avoidance strategies online, but it also improves obstacle-avoidance mapping relations offline through $\Delta O\left(t\right)$. So, Equation (1), considering incremental learning, can be improved as:(2)$$S\left(t+1\right)=f\left(\begin{array}{l} S\left(t\right),O\left(t\right),\Delta O\left(t\right),\\ OP\left(t\right),AS\left(t\right) \end{array}\right)$$

The improved structure can be described as in Figure 7. The final goal is to avoid obstacles immediately online and to find optimal paths offline. The basic algorithm is an artificial potential method, which is an algorithmic approach for collision avoidance that assigns potential values to different positions in the environment based on their proximity to obstacles. The core idea is to create a field where the UAV experiences a force that is inversely proportional to the distance from the obstacle. This force acts as a repulsive effect that guides the UAV away from obstacles. By calculating the gradient of the potential field, the UAV can determine the direction of the safest path, which is typically the path of least resistance or of lowest potential value. This method is efficient because it allows for real-time navigation decisions without the need for complex computations, making it suitable for dynamic environments where quick responses are necessary.

### 3.2. Construction of an Avoidance Strategy and an Obstacle Pattern

Regardless of a quad-rotor’s rotation around the center of mass, it can be described as a controllable mass. The gridding model of a UAV is constructed as shown in Figure 8, in which a UAV has eight flyable directions.

An avoidance strategy can be obtained via the sequential combination of flying actions:(3)$$A=\left\{\left.a_{1},a_{2},\cdots ,a_{n}\right|a_{i}\in Actions\right\}$$

Avoidance-strategy cognition is the combination of well-organized avoidance strategies:(4)$$AB=\left\{AS_{1},AS_{2},\cdots ,AS_{k}\right\}$$

Suppose the k-th obstacle’s position $\left(\rho_{k},\phi_{k}\right)$ and velocity $\left(v_{k},\theta_{k}\right)$ are given in a polar coordinate system, so the definition of the k-th obstacle can be described as
(5)$$ob_{k}=\left(\rho_{k},\phi_{k},v_{k},\theta_{k}\right)$$

In order to illustrate the computational process, Equation (5) is simplified to Equation (6), where $A_{1}$ to $A_{4}$ means $\rho_{k}$, $\phi_{k}$, $v_{k}$, $\theta_{k}$, respectively.
(6)$$ob=\left(A_{1},A_{2},A_{3},A_{4}\right)$$

For the determined value $x_{0}$ of the q-th property, a subordinating degree function $A_{qp}\left(x_{0}\right)$ can be obtained by the maximum-membership-degree principle. The fuzzy sets corresponding to $x_{0}$ are defined in Equation (7), where $x_{0}\in A_{qp}\;(1\leq p\leq N)$.
(7)$$A_{qp}\left(x_{0}\right)=\vee_{q=1}^{N}\left\{A_{qp}\left(x_{0}\right)\right\}$$

The osculating function $\sigma_{i}$ is introduced to compare whether obstacle k and obstacle l belong to same obstacle pattern.

Supposing that $\sigma_{i}$ is based on the i-th property,
(8)$$\sigma_{i}\left(A_{ik},A_{il}\right)=\frac{1}{2}\left[A_{ik}\cdot A_{il}+\left(1-A_{ik}\times A_{il}\right)\right]$$
where $A_{ik}$ and $A_{il}$ mean the fuzzy sets based on the i-th property of obstacle k and l, respectively, and $A_{ik}\cdot A_{il}$ and $A_{ik}\times A_{il}$ are the inner and outer products.

There are many obstacles in an environment, so we introduce an obstacle pattern to memorize obstacles as $OP={\left[ob_{1}^{\prime },ob_{2}^{\prime },\cdots ,ob_{k}^{\prime }\right]}^{\prime }$, and the obstacle pattern cognition is a matrix set as $OB=\left\{OP_{1},OP_{2},\cdots ,OP_{k}\right\}$.

Suppose that the obstacle pattern around UAV is $P={\left[ob_{1}^{\prime },ob_{2}^{\prime },\cdots ,ob_{m}^{\prime }\right]}^{\prime }$, and the comparing obstacle pattern knowledge now is $Q={\left[ob_{1}^{\prime },ob_{2}^{\prime },\cdots ,ob_{n}^{\prime }\right]}^{\prime }$. To ensure that obstacles in Q can completely replace P, every obstacle $\left(ob_{k}\right)$ in P should find a high closed-degree function with $ob_{l}$ in Q, which means that $\forall ob_{k}\in P$, $\forall i\left(i=1,2,3,4\right)$, $\exists ob_{l}\in Q$:(9)$$\sigma_{i}\left(A_{ik},A_{il}\right)\geq c_{i}$$
where $c_{i}$ is the threshold value of the closed degree for the i-th property.

The similarity degree function $sim\left(P,Q\right)$ can be designed by assigning a different weight $\omega_{i}$ to the corresponding obstacle property:(10)$$sim\left(P,Q\right)=\sum_{i=1}^{4}\sigma_{i}\left(A_{ik},A_{il}\right)\times \omega_{i}$$

For the purpose of selecting the most similar obstacle pattern $Q_{0}$, the obstacle pattern P is compared with all the obstacle patterns so that the optimal similar mapping pattern $Q_{0}$ can be chosen:(11)$$if\underset{\forall Q_{k}\in TB}{\max}\left(sim\left(P,Q_{k}\right)\right)\;,\;Q_{0}=Q_{k}$$

Finally, the knowledge incremental function $f_{add}$ is constructed so as to judge whether or not to add a new obstacle-avoidance mapping relation to the memories:(12)$$f_{add}=\left\{\begin{array}{l} 1\;,\;if\;sim\left(P,Q\right)\geq c_{sim}\\ 0\;,\;if\;sim\left(P,Q\right)<c_{sim} \end{array}\right.$$
where $c_{sim}$ is the threshold of similarity function; the new obstacle pattern can be described and represented if $sim\left(P,Q\right)\geq c_{sim}$.

## 4. Simulation and Analysis

To compare the effectiveness and capability of our method, two representative algorithms were introduced (ant colony (AC) algorithm [48] and cognitive game (CG) algorithm [49]).

The AC algorithm is a simulated evolutionary algorithm derived from the biological world. It has the characteristics of distributed computing, positive feedback of information, and heuristic searching. It is one of the most effective algorithms for solving the problem of UAV path planning.

On the basis of the autonomous anti-collision responses of UAVs in a non-isolated airspace, the cognitive game algorithm transforms the collision-avoidance problem into a game model. By establishing a kinematics model of UAVs and obstacles, the CG algorithm puts forward a solution method for safe path planning.

All simulations were run under the Matlab 12.0, 3.3 GHz Inter Core i3, Windows XP system. Assume that the UAV’s detection range is 10 m and its flight speed is 2 m/s, the training scenarios involve obstacles randomly occurring within the UAV’s detection range. As shown in Figure 9a–l, 12 diverse training samples were constructed and safe avoidance strategies were obtained by a basic anti-collision algorithm. In the diagram, the black areas are designated to represent obstacles, which could be buildings, trees, or any other physical entities that may impede the flight of a drone. The blue lines delineate the collision-avoidance flight paths that the drone takes to navigate around these obstacles. The gray portions symbolize the drone itself. With this color-coding, it is evident how the drone utilizes its sensors and algorithms to intelligently plan a safe flight path based on the surrounding obstacle information. This not only showcases the drone’s autonomous navigation capabilities but also highlights its adaptability and flexibility in complex environments, successfully finding and executing an escape route from densely populated obstacle areas.

The relation between cognitive learning knowledge and the training samples after training is shown in Figure 10.

We can infer that the number of training samples is unequal to the amount of knowledge, because we introduced the concept of fuzzy matching, which combines similar obstacles and avoidance strategies so as to reduce dimensionality. The similarity-degree value of obstacles in Figure 9e,f is 0.87 (the value between the avoidance strategies in Figure 9d,e is 0.92), which means that we can use the same obstacle pattern (and avoidance strategy) to classify them. But the similarity-degree value of the obstacle patterns between Figure 9c,l is 0.3, so we have to construct two new obstacle patterns.

The environment is a 200 m × 200 m square crowded with 10 obstacles. A UAV has to fly from the start position (0,0), pass by two targets at position (60,60) and (140,140), and finally end at position (200,200).

The results are shown in Figure 11a and Figure 12a, where the gray circles are the amplified regions of three flying moments. To illustrate the safety degree of a UAV, an equivalent distance function is constructed. Supposing that there are k obstacles within sensory range, then the equivalent distance function can be defined as shown in Equation (13):(13)$$f_{ed}=\frac{\rho_{1}}{R}\cdot \frac{\rho_{2}}{R}\cdots \frac{\rho_{k}}{R}=\prod_{i=1}^{k}\frac{\rho_{i}}{R},\;\rho_{i}\in \left(0,R\right)$$
where R is the sensory range, and ρ is the distance between each obstacle and the UAV in R.

The results for the $f_{ed}$ are shown in Figure 11b and Figure 12b. We can infer from Figure 11a and Figure 12a that the AC algorithm and the CG algorithm cannot deal with complex obstacles. In Figure 11a, though the AC algorithm can obtain a safe path at the beginning, when it encounters many obstacles, the UAV cannot keep a safe distance from the crowded obstacles (see the local magnification of the black path). In Figure 12a, the cognitive game algorithm performs better than the ant colony algorithm, however, the UAV has a collision with obstacles midway ($f_{ed}=0$). The method of the ant colony and cognitive game algorithms (see Figure 11b and Figure 12b) involves collisions with obstacles at moments 86.7 s and 152.6 s (when the value of $f_{ed}$ decreases to 0). In contrast, the proposed method can obtain a safe path and keep the $f_{ed}$ at the safe value.

This is because the ant colony algorithm requires a continuously re-planned flight path, and the cognitive game algorithm’s computing time is too long. However, our method can obtain an anti-collision path directly and dispense with repetitive calculations for similar obstacle patterns. When encountering complex obstacles, a cognitive UAV will find similar obstacle patterns based on the similarity-degree function, and then the corresponding avoidance strategy can be obtained. If there is no similar obstacle pattern, a new path can be designed, so the UAV can develop itself incrementally.

To make the results more convincing, the three algorithms were compared in the same environment repeatedly. Table 2 shows the statistical minimum value of the equivalent distance function $Min_{f}$ and the corresponding time $t_{\min}$. We find that AC and CG often result in a collision in crowded environment, because there always exists an $Min_{f}=0$. As assumed above, $Min_{f}$ is the minimum value of $f_{ed}$; when it equals 0, a collision occurs. In repeated results using CBN, the value of $Min_{f}$ is always positive, which means that this proposed method always keeps a safe distance between obstacles and the UAV.

Finally, fast anti-collision is achieved like that of humans’ conditioned reflex. The effectiveness of the anti-collision algorithm is improved by avoiding duplicative planning in a similar environment.

## 5. Experiment and Analysis

As shown in Figure 13, the UAV’s perception model has eight directions. When the distance between the UAV and the obstacles is less than the threshold value, there is an obstacle occlusion in this direction.

The selected equipment and materials of the UAV are shown in Figure 14.

Figure 15 demonstrates the anti-collision experiment utilizing an unmanned aerial vehicle (UAV) in the presence of an obstacle in a single direction, where the red arrows represents the moving direction. It is observable that as the dynamic obstacle (people holding a planar board close to the UAV as the dynamic obstacle) approaches, the UAV is capable of swiftly maneuvering away from the obstacle and maintaining a safe distance. The upper right corner of the image illustrates the UAV’s trajectory in response to the obstacle’s movement. It can be discerned that the UAV consistently self-regulates to maintain a safe distance while flying, contingent upon the proximity of the obstacle to itself.

Table 3 shows the time comparison between the traditional anti-collision method and the proposed method. It can be seen that the traditional method needs more time from perceiving the obstacles to finally completing the anti-collision behavior because it needs to solve the anti-collision algorithm, convert the calculated anti-collision strategy into angle and throttle signals, and then convert it into control commands. However, the conditioned reflection method is directly mapped to the anti-collision instructions, which saves on the intermediate calculation steps, speeds up the operation time of the algorithm, and improves the anti-collision efficiency.

In addition to the practical testing of the UAV in a single-obstacle environment, this paper also conducted anti-collision experiments with obstacles in two different directions As shown in Figure 16, one was an anti-collision test in the constructed right-angle-obstacle environment, and Figure 17 features an anti-collision test in the constructed horizontal-obstacle environment, where the red arrows represents the moving direction. It can be observed that the UAV can quickly respond to these obstacles and can maintain its stability in a safe area. In Figure 16 and Figure 17, annotations made in the upper right corner indicate the UAV’s movement trend in response to the obstacles. From the top-down view, it is evident that when the UAV is too close to an obstacle, it exhibits a repulsive behavior, and when it is too far away, it exhibits an attractive behavior, thus maintaining a safe distance.

## 6. Conclusions

In terms of UAVs, we have argued that it can follow a cognitive method like the human conditioned reflex. After reviewing anti-collision studies, a cognitive brain-like model for UAVs has been proposed from the context of a bio-behavioral perspective. Following are some concluding remarks: Unlike traditional anti-collision algorithms, the proposed method can cognize by interacting with environment gradually. And it can learn new anti-collision mapping relations by online simulations. Similar to the human conditioned reflex, after experiencing similar obstacles, cognitive UAVs have the ability to avoid obstacles without thinking.

Still, there are several issues in need of attention and further investigations, including practical studies and algorithm optimization.

## Figures and Tables

**Figure 1 sensors-24-02790-f001:**
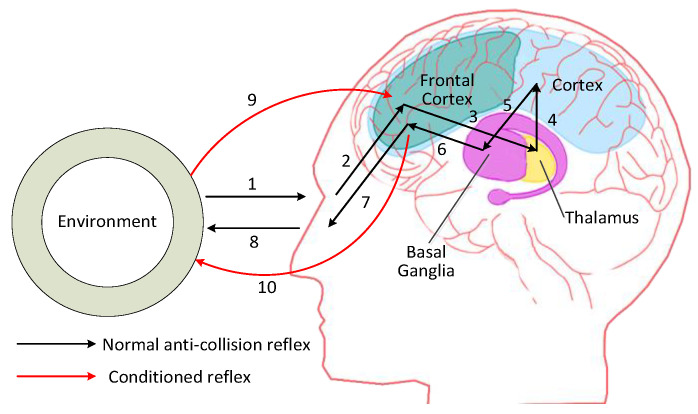
Brain pathway of the conditioned reflex cycle in the human anti-collision response.

**Figure 2 sensors-24-02790-f002:**
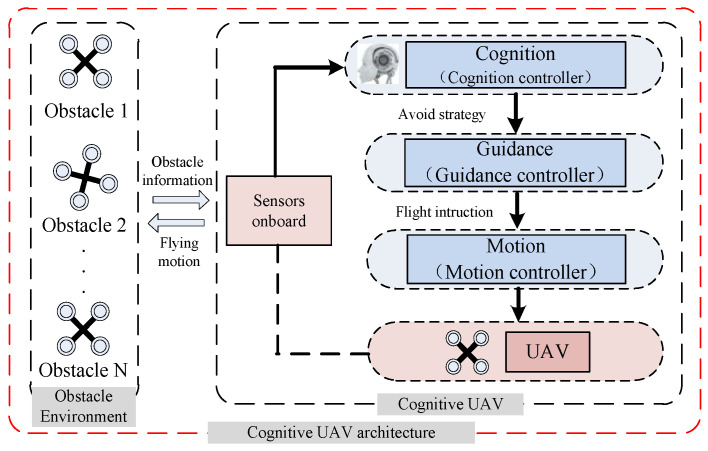
The cognitive control architecture of a UAV.

**Figure 3 sensors-24-02790-f003:**
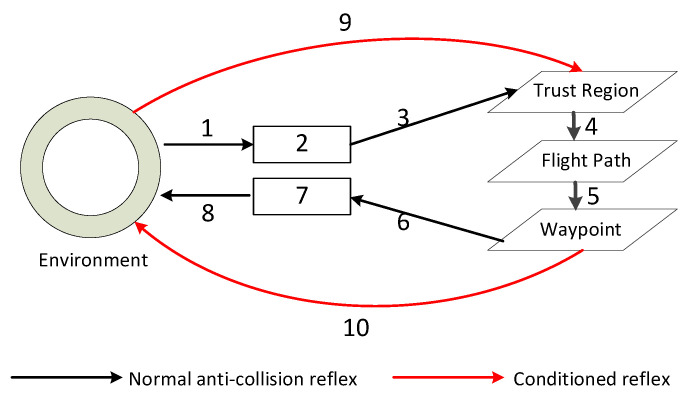
The architecture of the RBN for an anti-collision response.

**Figure 4 sensors-24-02790-f004:**
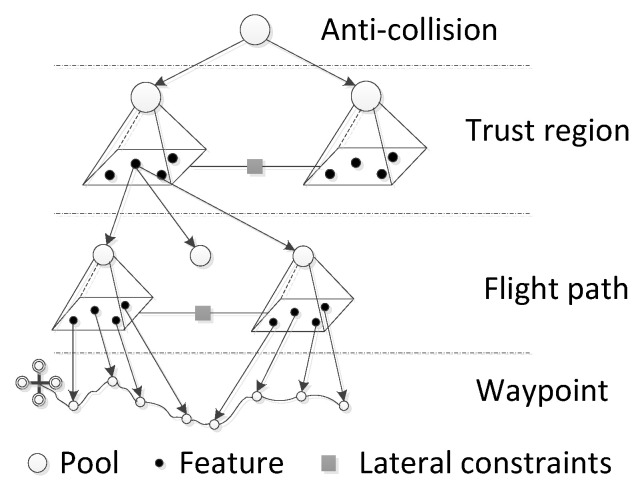
The hierarchical RBN model of the trust region, flight path, and waypoints.

**Figure 5 sensors-24-02790-f005:**
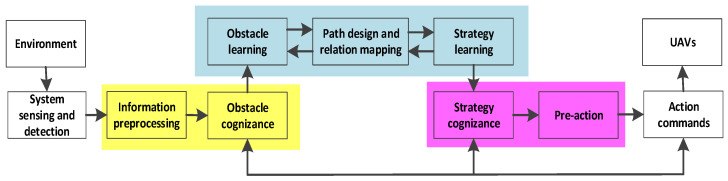
A brain-like conditioned reflex cycle in cognitive UAVs.

**Figure 6 sensors-24-02790-f006:**
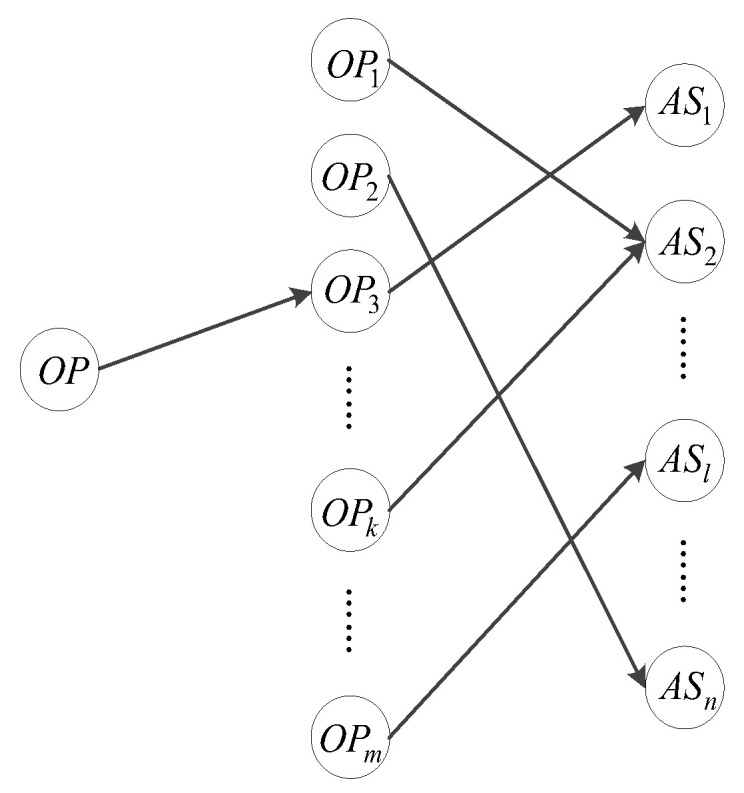
Structure of obstacle-avoidance mapping.

**Figure 7 sensors-24-02790-f007:**
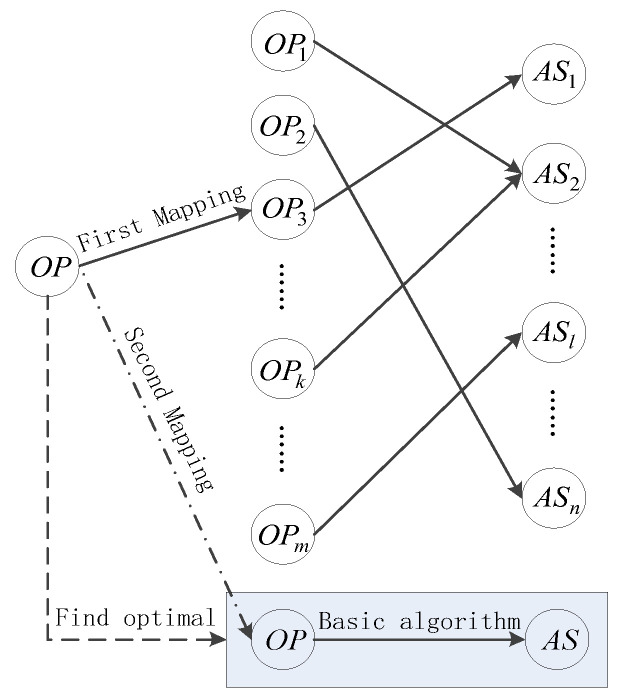
Improved structure of obstacle-avoidance mapping.

**Figure 8 sensors-24-02790-f008:**
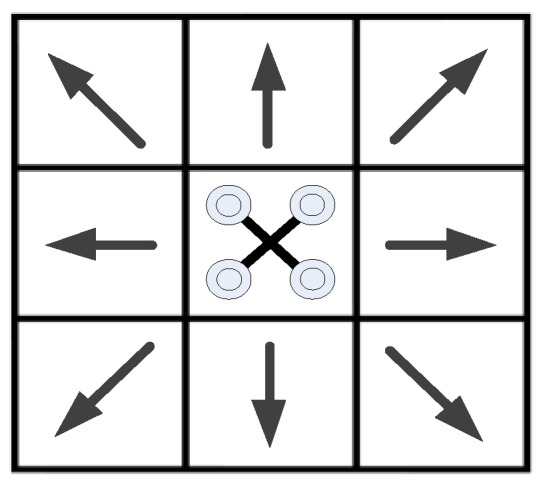
Gridding model of a UAV.

**Figure 9 sensors-24-02790-f009:**
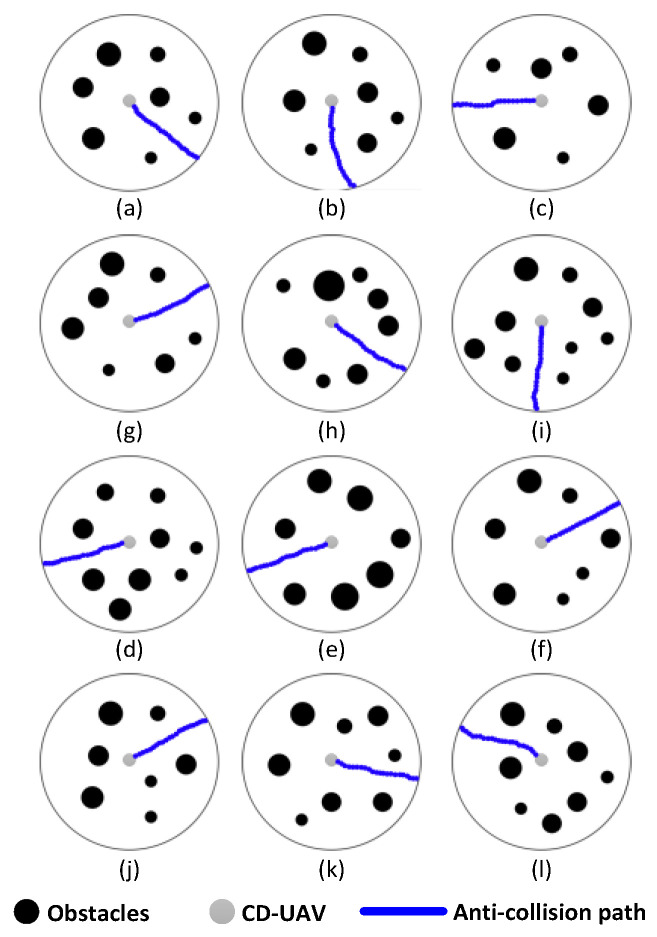
The 12 diverse training samples.

**Figure 10 sensors-24-02790-f010:**
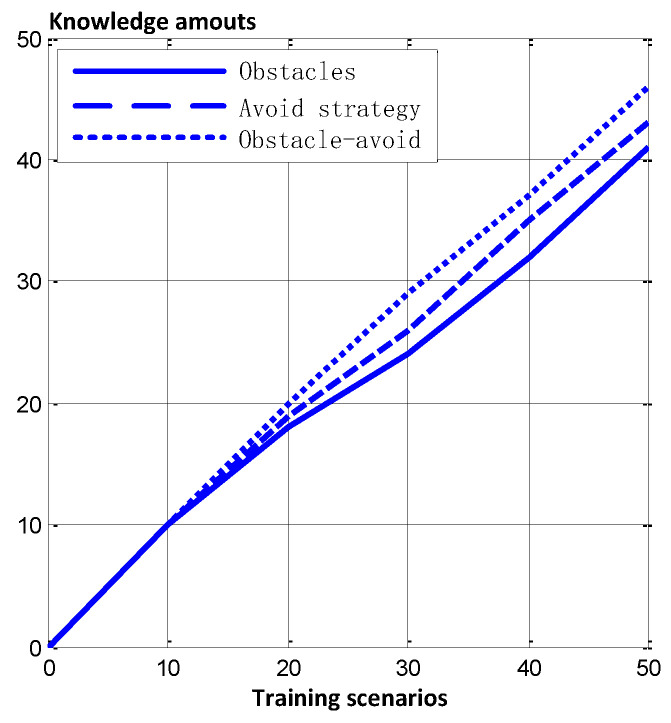
The relation between developmental knowledge and the training samples.

**Figure 11 sensors-24-02790-f011:**
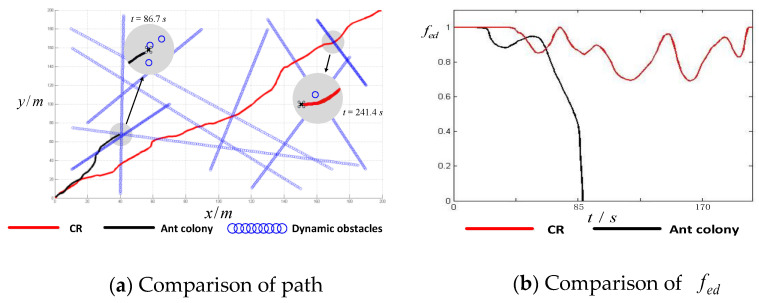
Comparison of CR and AC among crowded obstacles.

**Figure 12 sensors-24-02790-f012:**
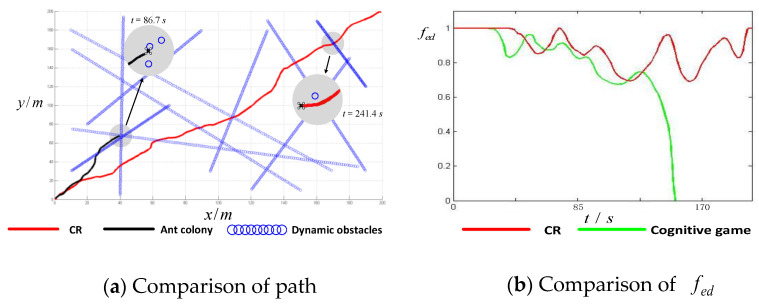
Comparison of CR and CG among crowded obstacles.

**Figure 13 sensors-24-02790-f013:**
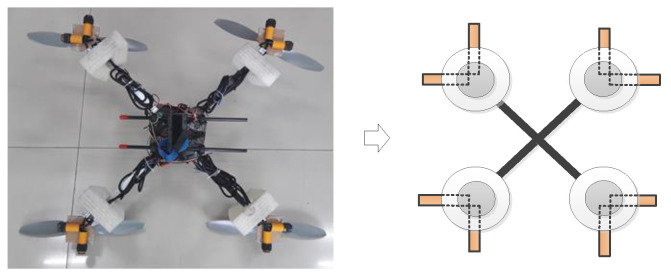
The UAV’s perception model.

**Figure 14 sensors-24-02790-f014:**
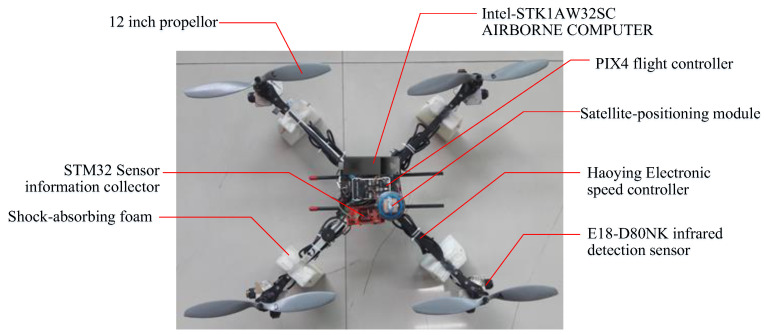
The selected equipment and materials of the UAV.

**Figure 15 sensors-24-02790-f015:**
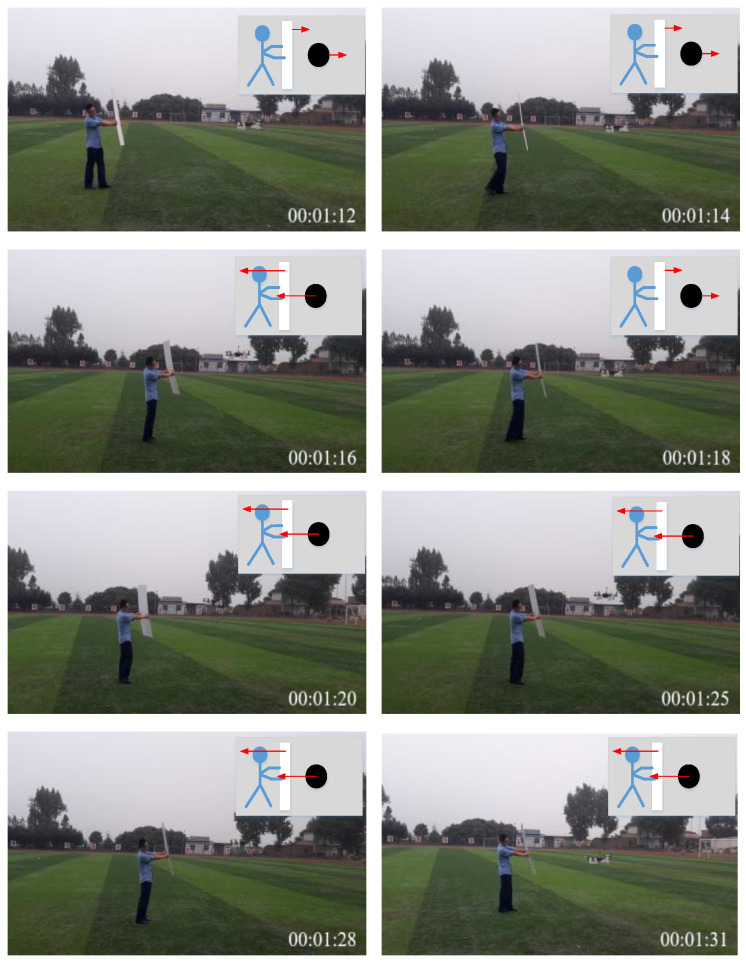
A UAV’s anti-collision response to one obstacle.

**Figure 16 sensors-24-02790-f016:**
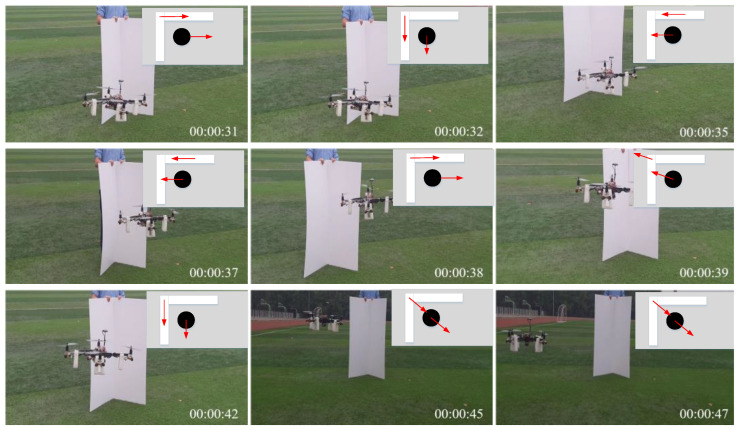
Anti-collision test in the constructed right-angle-obstacle environment.

**Figure 17 sensors-24-02790-f017:**
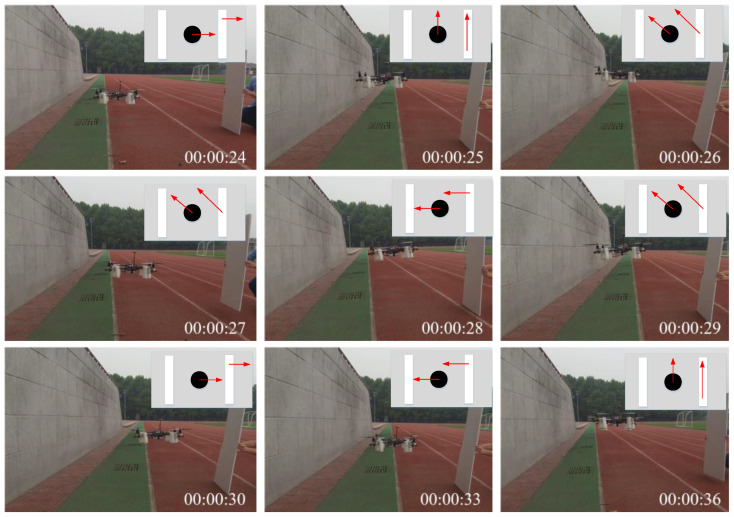
Anti-collision test in the constructed horizontal-obstacle environment.

**Table 1 sensors-24-02790-t001:** Detailed description of the eight processes.

Number of Processes		Functions
1	Sensing	UAVs use sensors onboard to detect obstacles around them.
2	Preprocessing	To preprocess the mass and unclassified information of obstacles.
3	Cognition	In collision avoidance, this process involves having an awareness of familiar environment and cognizing obstacles.
4	Obstacle learning	This process aims to learn and reserve paths and to map the paths between trust regions and waypoints.
5	Behavior	Its function is to design an anti-collision strategy and to behave safely in an environment.
6	Strategy learning	This process is to memorize anti-collision strategies.
7	Pre-action	The waypoints cannot be decoded and acted by UAVs, so this process clarifies them so as to transform them into commands.
8	Action	Cognitive UAVs follow the commands from the RBN to avoid crowded obstacles.

**Table 2 sensors-24-02790-t002:** Repeated simulation results.

Test No.	AC		CG		CR	
	Min_f_/(m)	t_min_/(s)	Min_f_/(m)	t_min_/(s)	Min_f_/(m)	t_min_/(s)
1	0	96.7	0	169.2	0.25	76.8
2	0	87.5	0	89.4	0.42	89.1
3	0	165.2	0	164.3	0.34	79.7
4	0	148.3	0	172.8	0.43	267.5
5	0	87.9	0	94.6	0.26	285.6

**Table 3 sensors-24-02790-t003:** Time comparison between the traditional anti-collision method and conditioned reflection method.

Environment	1st	2nd	3rd	4th
Traditional method	6.4 s	8.6 s	7.1 s	6.5 s
Conditioned reflection	3.7 s	5.2 s	3.6 s	4.4 s

## Data Availability

All data generated or analyzed during this study are included in the manuscript. All data included in this study are available upon request by contact with the corresponding author.

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
