# Peer review of "Cognitive Control Architecture for the Practical Realization of UAV Collision Avoidance"

_sensors, 2024, doi:10.3390/s24092790_

Round 1
Reviewer 1 Report
Comments and Suggestions for Authors
The paper provided an interesting topic that described the brain science asscociated with anti-collision UAV.
Some comments:
1. It's difficult to understand Figure 3 and Figure 5. Event-state or processess should use suitable shapes. There are lines(black or red) (eg. 1,3) and rectangles (2,7) to depict processes equally, WHY? Same problem for Figure 5 and corresponding sentencs.
2. It would be better if there are topviews for Figure 15, 16 and 17 that the readers can easily find the horizonal location and changes.
Comments on the Quality of English LanguageModerate editing of English language required
Reviewer 2 Report
Comments and Suggestions for Authors
This paper provides a substantial contribution to the field of UAV navigation and control systems. Its strengths lie in its innovative integration of cognitive science principles with robotic control. However, there are still some points needed to be addressed before accepted.
11. In introduction, Line 22 to 32 can combine to one paragraph instead of separate them.
22. In the literature review, the authors can provide a more specific results of simulation and experiments to explain the research gap that this paper aims to fill.
33. From line 61-65, prefer not to say the organization of paper and just talk about the main conclusions and contributions of this paper.
44. In part 2, the brain pathway in human brains is somehow over detailed and it can be concise. If the authors can expand the conditioned reflex(CR) mechanism and how it integrates with UAV control, it will be more helpful. So, I suggest you combine part 2 and part 3.
55. In line 163, the author explains that it can enrich itself incrementally. Could you explain it ?
66. For RBN, does this method propose in this paper have a relationship with machine learning?
77. In table 1 last process, the author proposed that “Cognitive UAVs follow the commands from RBN to avoid crowded obstacles.”, could you pls describe how the RBN constructed and executed with more details?
88. In line 185, the author explains that “Flight path is obtained by calculations”, how is this calculation conducted and where is calculation done, such as trust region, etc?
99. In line 223-224, the authors provide that the “find the optimal paths of offline”, what does mean offline, without calculation or any other? Also, In Fig. 7 , the author labels the basic algorithm, could you describe this basic algorithm in more detail?
110. In Line 240, the authors use “maximum membership degree principle”. Pls describe how this is applied in this control, for example, how to define the fuzzification and Rule evaluation?
111. In line 254, the author uses the threshold value of close degree, could you describe how to define it?
112. In equation 10, how to define the weight and describe the criteria of choosing weight. In Deep learning, weight is calculated through the learning, how about this one?
113. In fig. 9, how to choose these 12 samples and I think this important since it is critical to train the model.
114. In conclusion, can you combine the four paragraphs to two, one for the conclusion and advantage over the previous method, another one is for the future plan? This will make the description more concise.
Round 2
Reviewer 2 Report
Comments and Suggestions for Authors
I have no comments